# The Effect of the Cooling Rates on the Microstructure and High-Temperature Mechanical Properties of a Nickel-Based Single Crystal Superalloy

**DOI:** 10.3390/ma13194256

**Published:** 2020-09-24

**Authors:** Xiao-Yan Wang, Meng Li, Zhi-Xun Wen

**Affiliations:** 1School of Mechanics, Civil Engineering and Architecture, Northwestern Polytechnical University, Xi’an 710072, China; mengli1377@mail.nwpu.edu.cn (M.L.); zxwen@nwpu.edu.cn (Z.-X.W.); 2School of Science, Xi’an Shiyou University, Xi’an 710065, China

**Keywords:** nickel-based single-crystal superalloy, solution heat treatment, cooling rate, high-temperature mechanical properties, microstructure

## Abstract

The as-cast alloy of nickel-based single-crystal superalloy was used as the research object. After four hours of solution treatment at 1315 °C, four cooling rates (water cooling (WC), air cooling (AC) and furnace cooling (FC1/FC2)) were used to reduce the alloy to room temperature. Four different microstructures of nickel-based superalloy material were prepared. A high-temperature tensile test at 980 °C was carried out to study the influence of various rates on the formation of the material’s microstructure and to further obtain the influence of different microstructures on the high-temperature mechanical properties of the materials. The results show that an increase of cooling rate resulted in a larger γ′ phase nucleation rate, formation of a smaller γ′ phase and a greater number. When air cooling was used, the uniformity of the γ′ phase and the coherence relationship between the γ′ phase and the γ phase were the best. At the same time, the test alloy had the best high-temperature tensile properties, and the material showed a certain degree of plasticity. TEM test results showed that the test alloy mainly blocked dislocations from traveling in the material through the strengthening effect of γ′, and that AC had the strongest hindering effect on γ′ dislocation movement.

## 1. Introduction

Because of its excellent mechanical properties at high-temperature, nickel-based single-crystal superalloy is widely used in aero-engine hot-end parts and has become the preferred material for manufacturing turbine blades [1,2]. Through solution and aging treatments, the alloy makes the precipitation phase γ′ present a highly regular cubic structure and is uniformly dispersed in the matrix phase γ, which greatly improves the high-temperature mechanical properties of the material [2,3]. Therefore, the mechanical properties of superalloys are significantly affected by the final microstructures.

In order to obtain the appropriate microstructures of nickel-based superalloys, an adequate heat treatment process is very important. In the past, some studies have been conducted to examine the microstructural evolution of nickel-based superalloys in different heat treatment processes [3,4,5,6]. Grosdidier [7] studied the precipitation and dissolution of the γ-phase in nickel-based alloys AM1 and CMSX-2 during heat treatment, discussed the microstructural evolution of the precipitation phase just after precipitation and proposed that the precipitation phase would undergo deformation from spherical to cubic to dendritic from nucleation to growth. You et al. [8] studied the effect of solid solution temperature on the precipitation and strengthening mechanisms of the superalloy Inconel 718. It was found that the γ′ phase precipitates at a higher temperature than the γ′′ phase and is distributed in the matrix at a certain volume fraction at 1150 °C. Radis et al. [9] studied the microstructure and distribution of γ′ phase precipitation in the continuous cooling process of the nickel-based superalloy UDIMET 720Li after solution treatment and proposed a kinetic model for the precipitation of the strengthening phase during the cooling process. Mostafaei et al. [10] studied the effect of solid solution and aging on the microstructure and mechanical properties of superalloy 625. Wang [11] mainly explored the effect of the solution temperature and holding time on the γ′ phase dissolution behavior of IN100 and DS Rene125 superalloys during the solution treatment process and explored the kinetic factors that strengthen the phase dissolution behavior during the solution treatment process. Mitchell et al. [12] discussed the relationship between cooling rate, microstructure, mismatch and hardness of nickel-based superalloys. It is clear that the reasonable heat treatment process has an important influence on the formation of appropriate micro microstructure of superalloy.

Precipitation-strengthened superalloys obtain a supersaturated solid solution through solution heat treatment, which provides the best conditions for subsequent aging treatment. This plays a vital role in the heat treatment system. In solution heat treatment, temperature, holding time and cooling rate are the key factors affecting the heat treatment results. The effect of the cooling rate on the microstructure and mechanical properties of superalloys during heat treatment has been discussed in the past [12,13,14,15,16,17,18]. Recently, Letniko et al. [19] studied the effect of quenching rate on the microstructure and mechanical properties of cast superalloy VZh175-ID and calculated the relationship between the cooling rate and the average size of the γ′ phase and the strength of the material. Wu et al. [20] studied the effects of single-step and multistep heat treatment on the mechanical properties of powdered superalloys. The cooling rate plays an important role in tailoring the microstructure, size and size distribution of γ′ precipitates, which is clarified by the precipitation calculation enabled by multicomponent thermodynamic database. Lin et al. [21] studied the effects of interrupted and direct water-cooled methods on the microscopic microstructure evolution, mechanical properties and fracture mechanism of nickel-based superalloys.

In summary, for nickel-based single-crystal superalloys, the cooling rate of solution heat treatment can significantly affect the microstructure such as the size of γ′ phase and the width of γ phase channel and then have a significant impact on the mechanical properties. Therefore, in order to study the effect of different solution cooling rates on the microstructure and mechanical properties of the tested nickel-based single-crystal superalloy, different solution cooling rates were used, and high-temperature mechanical properties were determined.

## 2. Test Materials and Process

The materials used in this study were as-cast alloys of experimental nickel-based single-crystal superalloys. The nominal element content is shown in Table 1. First, pure raw materials were used to melt the master alloy in a vacuum induction melting furnace (Shenyang Hengrun Vacuum Technology Co., Ltd., Shenyang, China), and then a single-crystal test bar with [001] orientation was cast in a high-gradient directional solidification vacuum furnace using a spiral crystal selection method. The crystal orientation of the test bar was determined by Laue’s method. The deviation between the [001] crystal orientation of the bar and the stress principal axis was kept within 15 degrees. The test bar with a deviation within 10 degrees was selected as the experimental material of this study.

The as-cast single-crystal test bar was heated in a stepwise manner in a smart multi-stage temperature-controlled vacuum high-temperature tube furnace to its solution temperature of 1315 °C, followed by 4 h solution heat treatment. This was followed by furnace cooling 1 (0.15 °C)/s), furnace cooling 2 (0.6 °C/s), air cooling (72 °C/s) and water cooling (138 °C/s) [22], as four different cooling rates to make the material cool to room temperature. For the convenience of marking, FC1, FC2, AC and WC are used to represent four different cooling methods in the following text. The temperature control accuracy of vacuum high-temperature tube furnace was 5 °C. During the test, a B couple was inserted into the uniform temperature zone near the test bar of the vacuum high-temperature tube furnace to observe the temperature change in real time. The temperature–time curve during the test is shown in Figure 1. Test blocks (3 × 5 × 5 mm) were cut from the alloys with four different heat-treatment methods for microstructure observation. The observation surface was perpendicular to the alloy’s [001] orientation. Metallographic specimens were prepared for observation by waterproof abrasive paper of 800#, 1200# and 2000#, respectively, and then polished and chemically etched. The composition of the etchant used was 20 g CuSO_4_ ·5H_2_O + 5 mL H_2_SO_4_ + 50 mL HCl+100 mL H_2_O.

Field emission scanning electron microscope (SEM) (Gemini300, Carl Zeiss AG, Oberkochen, Germany) (ZEISS Gemini300) was used to observe the microstructure of the alloy after different heat-treatment methods. Image-J software was used to statistically analyze the alloy’s γ′ phase size, γ phase channel width and their microstructure and distribution.

The alloys with four different heat-treatment methods were made into I-shaped flat specimens along the [001] orientation. The dimensions of the I-shaped specimens are shown in Figure 2. The surface of the sample was polished to the same roughness with 800# sandpaper to eliminate the influence of processing defects and residual stress. The I-shaped specimens with different heat-treatment methods were subjected to high-temperature mechanical tensile tests at 980 °C with a customized high-temperature creep machine (RDL100, China Machinery Experimental Equipment Co., Ltd., Changchun, China). The reference standard for this test was the metal high-temperature tensile test standard GBT 228.2–2015. The selection of the tensile temperature was set according to the common working temperature of the test alloy in actual engineering applications. The high-temperature tensile test sample was not a standard test piece, so it was clamped by a designed special fixture. As shown in Figure 2, the test piece was held on a creep testing machine with an annular heating high-temperature furnace through a self-made fixture. Then, bind 3 S-type thermocouples were attached to the upper, middle and lower sections of the sample to ensure temperature control during the test. Finally, after the furnace was heated to 980 °C for 30 min, a tensile test was performed. Displacement control was adopted, and the tensile rate was 0.2 mm/min. The mechanical properties such as yield strength (σ_s_/σ0.2), ultimate tensile strength (σ_b_), elastic modulus (E), elongation (δ) and reduction of area (ψ) were obtained. The elastic modulus (E) was measured with a high-temperature elastic modulus tester (IET-1400VP, Luoyang Zhuosheng Testing Instrument Co., Ltd., Luoyang, China).The displacement control was adopted, and the displacement change rate was 0.2 mm/min. During the experiment, three S-type thermocouples were used to measure the experimental temperature in the upper, middle and lower parts of the sample, and the experimental temperature difference was guaranteed to be within 5 °C. In order to reduce the error caused by the equipment, the samples of each heat-treatment method were repeatedly subjected to three high-temperature tensile experiments, and all high-temperature tensile experiments were completed on the same creep testing machine.

The ultra-depth-of-field optical microscope (OM) (VHX-6000, KEYENCE, Osaka, Japan) was used to observe and analyze the fracture of the samples after high-temperature tensile test. A transmission electron microscope (TEM) (Talos F200X G2, FEI, Hillsboro, WA, USA) was used to observe the microstructure of the fracture side ([100]) of the samples after high-temperature tensile test and analyze its strengthening and failure mechanism.

## 3. Test Results and Discussion

### 3.1. Microstructure After Heat Treatment

The SEM microstructure after four different heat-treatment methods is shown in Figure 3. The white part is the matrix γ phase, and the black part is the strengthened γ′ phase chemically corroded. There are almost no carbides. The microstructure is mainly composed of γ nickel-based solid solution containing γ′ phase and a small amount of γ + γ′ eutectic structure. It can be clearly seen from Figure 3 that the cooling rate after solution heat treatment had a significant impact on the shape, size and distribution of the precipitated γ′ phase. As the cooling rate increased, the size of the γ′ phase was significantly smaller and more uniform. In addition, as the cooling rate increased, the shape of the γ′ phase changes from a cube to an irregular curved surface. When air cooling was used, the shape of the γ′ phase was already spherical. When the cooling rate was slower, the atoms diffused more fully in the alloy, and the γ′ phase had time to grow up and merge into a larger size γ phase.

We analyzed the equivalent side length of the γ′ phase and the width of the matrix-phase channel in the SEM image by using image analysis software. The results are shown in Table 2. It can be seen from Figure 3a that when the cooling rate was FC1, the γ′ phase presented a relatively regular cube, but the size of the γ′ phase was different and the degree of homogenization was poor. The equivalent side length of the γ′ phase was about 375 nm, the arrangement was relatively orderly, and the width of the matrix-phase channel was about 30 nm. The shape of the γ′ phase cube did not change much. In the dendrite nucleus area, there was a tendency to form a butterfly-like structure consisting of four strengthening phases. When the cooling rate was FC2, as shown in Figure 3b, the γ′ phase presents an irregular square shape, and the contour of the γ′ phase had a tendency to change to an irregular curved surface. The equivalent edge of γ′ phase was 183 nm, and the arrangement was relatively regular. The matrix-phase channel was obvious, the channel width was approximately equal to 15 nm, and a small amount of secondary precipitated phase was precipitated in the channel. When air cooled, as shown in Figure 3c, the shape of the γ′ phase was almost circular, and the equivalent side length of the γ′ phase was about 86 nm. The width of the matrix-phase channel was approximately 8 nm. The fine secondary precipitated phases in the channel increase significantly; Compared with the furnace cooled condition, the number of γ′ phases significantly increased and the volume reduced. When water cooling was used, as shown in Figure 3d, the γ′ phase was very small, and the shape was irregularly dotted toward a circle. The equivalent side length was less than 20 nm. The width of the matrix-phase channel was less than 1 nm. The γ′ phase was compact and disorderly, and the matrix-phase channel was very narrow. The channel was mixed with smaller secondary precipitated. It is difficult to clearly display the microstructure map under the existing electron microscope conditions.

During the solution treatment, most of the γ′ phase redissolved into the matrix and precipitated in the subsequent cooling process. The difference in cooling rate had a significant effect on the precipitation microstructure of the γ′ phase. The larger the cooling rate, the larger the nucleation rate of the γ′ phase, and the more and less γ′ phase was formed. Conversely, the lower the cooling rate, the smaller the nucleation rate of the γ′ phase, and more γ′ phase was formed. In the past literature of nickel-based single-crystal microscopic morphology evolution, the elastic strain energy model (LSW) caused by element diffusion [24,25,26] and the interface energy model (TIDC) caused by interface energy [27] were proposed. It has also been proposed in the literature that the change of γ′ phase microstructure depends on the joint effect of strain energy and interface energy. The interface energy and lattice mismatch degree were not simply linear. While the lattice mismatch at room temperature was negative, its value was more negative at high-temperature. If the lattice mismatch degree was positive, its value decreased or became negative at high-temperature [23]. At the same time, the literature [27] also pointed out that the γ′ phase size distribution (PSD) was more consistent with the strain energy model in the early stage of heat treatment. As the heat treatment time increased, it fit better with the interface energy model. Compared with a cube, the sphere had a smaller interface energy, so the γ′ phase of a curved surface was beneficial to reduce the interface energy. In this study, when the temperature was 1315 °C and the cooling rate was slow (FC1\FC2), the material stayed at high-temperature for a longer time and the γ′ phase had more time to grow and coarsen through diffusion after nucleation. The influence of interface energy on its microstructure was greater. This led to a coarse cubic microstructure. On the contrary, when the cooling rate was fast (AC\WC), the super cooling condition promoted the nucleation of a large number of γ′ phases and the strain energy at high-temperature was the main factor of its microstructure. This led to its spherical microstructure. Therefore, as the cooling rate increased, the microscopic size of the γ′ phase decreased, and the microstructure of the γ′ phase transitioned from cubic to spherical.

### 3.2. High-temperature (980 °C) Tensile Test Results

The high-temperature tensile test was carried out at 980 °C, using displacement control, and the test speed was 0.02 mm/min. The stress–strain curve is shown in Figure 4. The high-temperature tensile properties of the alloys with different cooling rates of heat treatment are shown in Table 3. In the table, the yield strength (lower yield strength), ultimate tensile strength, elastic modulus, elongation and reduction of area of the alloy were calculated. It can be seen from the stress–strain curve that all the experimental groups exhibit good plasticity. When air and water cooling were used, the alloy exhibited an obvious yielding process, and the material had a longer period of elastic deformation. However, when two furnace cooling heat-treatment methods with different cooling rated were used, the material did not undergo a significant yield process. In this study, the nominal yield stress was used to express the yield index, that is, the corresponding stress when 0.2% plastic strain was generated, which was expressed by the symbol σ_0.2_. From Table 3, as the cooling rate increases, the yield strength (σ_s_/σ0.2), ultimate tensile strength (σ_b_) and elastic modulus (E) all show a trend of first increasing and then decreasing, as shown in Figure 5.It was found that the air cooling after solution heat treatment had the best high-temperature tensile mechanical properties of the test alloy. In addition, when the low cooling rate furnace cooling heat-treatment method was used, the elastic modulus E of the alloy was significantly lower than that of the higher cooling rate AC and WC.

After the high-temperature tensile fracture of the four samples, the contour map of the fracture microstructure is shown in Figure 6 (the height shown is 50% of the true height). It could be seen that when air cooling was used, the height difference of the fracture was the smallest. Both water cooling and FC2 had fractures with a large height difference and the fracture clearly showed the shear failure characteristics of 45. It can be seen from Table 3 that when water cooling was used, the elongation δ and the reduction of area Ψ were both small. It can be seen that the fracture was perpendicular to the force direction and the section was rough by observing the fracture after the sample was broken, so the material exhibited a certain degree of brittleness. The elongation δ and the reduction of area Ψ when air-cooled were both large and the mechanical properties of the test alloy were the best when air-cooled. Therefore, the air-cooled heat-treatment method was used to maintain a certain degree of plasticity while showing better strength.

The mechanical properties of the alloy were directly related to its microstructure [20,28]. The main strengthening mechanism of the nickel-based single-crystal superalloy used in this study was the cubic coherent composition of γ′ phase and γ phase. The highly regular cubic structure γ′ phase was uniformly dispersed in the γ phase, which greatly improved the mechanical properties of the alloy. In the coherence relationship, the size and cube shape of the γ′ phase and the channel width of the γ phase were important quantitative indicators. It can be seen from the discussion in 3.1 that as the cooling rate increased, the degree of cubing of the γ′ phase gradually decreased and the equivalent side length of the γ′ phase and the channel width of the γ phase decreased—both showing a monotonous decreasing trend. However, as the cooling rate increased, the mechanical properties of the alloy showed a trend of first increasing and then decreasing, which had a mathematical extreme value. It can be concluded that when a certain cooling rate was adopted—that is, the equivalent side length and cubic degree of the γ′ phase of the alloy and the channel width of the γ phase reach a certain value—the high-temperature mechanical properties of the test alloy were the best. The conclusion of this study is that when the AC is used, the equivalent side length of the spherical γ′ phase is 86 nm and the channel width of the γ phase is 8 nm, the experimental alloy achieves the best high-temperature mechanical properties.

### 3.3. Microstructure Analysis After Tensile Fracture

An I-shaped specimen fractured by tensile test was cut perpendicular to the gauge length at a distance of 2 mm from the fracture surface. TEM samples were prepared along the thickness direction and observed by TEM.

The experimental material was a standard two-phase alloy, which mainly blocks dislocations through the strengthening effect of the precipitated phase γ′. When the dislocation moves to the vicinity of the phase γ′, it needs to cut through the phase γ′ or bypass the matrix phase γ between the phases γ′ to move on. Figure 7a–c shows the TEM image of the side of the fracture surface of the tensile pattern when AC was used. It can be seen that the precipitation phase γ’ was approximately spherical, and a large number of dislocations were intertwined in the matrix-phase channel. At the same time, a small amount of dislocations cut the γ´phase, leaving some dislocations on the surface of the γ′ phase. As shown in Figure 7a,c, the dislocations plugged in the matrix-phase channel and cut into the γ′ phase are marked with arrows. According to the analysis in Section 3.1, it can be seen that the lattice mismatch between the γ′ phase and the γ phase was greater than that in the FC, which will cause the coherent strain between the two phases. A higher elastic stress field was caused around the γ′ phase. The movement of dislocations was hindered, and the dislocation packing of the matrix-phase channel was formed. Figure 8a shows a schematic diagram of dislocations passing through the matrix-phase channel when AC was used. Figure 7a–c shows a TEM image of the fracture side of the tensile pattern when AC was used. It can be seen that the microstructure was very different from that when FC1 was used. The size of the γ′ phase was significantly larger. There were not a large number of dislocations entangling in the matrix-phase channel, but more dislocations cut into the γ′ phase and leave entangling dislocations on the surface, as shown in Figure 7f. The precipitates in Figure 7d may be caused by material defects and internal oxidation. Figure 8b is a schematic diagram of dislocations passing through the matrix-phase channel when FC1 was used.

It can be seen from Figure 4 and Figure 5 in Section 3.2 that FC1 had the lowest yield strength and AC had the highest yield strength. Compared with the FC1 heat-treatment method, when the AC heat-treatment method was adopted, the size of the γ′ phase was smaller and the number of matrix-phase channels was larger. Due to the greater degree of lattice mismatch between the two phases, a higher elastic strain field was generated, which had a stronger hindering effect on the movement of dislocations, and the resistance to be overcome for dislocations to bow in the AC matrix-phase channel was greater.

## 4. Conclusions

The effects of different cooling rates (FC1 (0.15 °C/s), FC2 (0.60 °C/s), AC (72 °C/s), WC (138 °C/s)) on the microstructure and mechanical properties of nickel-based single-crystal alloys were studied. The main conclusions are as follows:(1)The difference in cooling rates during the solution treatment had a significant effect on the precipitation microstructure of the γ′ phase. As the cooling rate increases, the larger the nucleation rate of the γ’ phase, the smaller the size of the formed γ’ phase, the greater the number;(2)Due to the interaction between strain energy and interface energy, the cubic degree of the γ′ phase gradually deteriorated, transitioning from square to spherical. When air cooling was used, the uniformity of the γ′ phase was better and the coherent relationship between the γ′ phase and the γ phase was also better;(3)Compared with FC1, FC2, WC, when AC was used, the yield strength (σ_s_/σ0.2), ultimate tensile strength (σ_b_) and elastic modulus (E) of the test alloy at high-temperature (980 °C) were the largest. In other words, the high-temperature tensile performance was the best under this condition. Combined with its microstructure, the size of the γ′ phase and the coherence relationship between the γ′ phase and the γ phase were the most reasonable;(4)When AC was used, the higher cooling rate increased the lattice mismatch of the material, resulting in coherent strain between the γ′ phase and the γ phase, which caused a higher elastic stress field around the γ′ phase and hinder dislocation movement. In addition, the smaller γ′ phase increased the proportion of the matrix-phase channel and had a stronger hindering effect on the movement of dislocations.

## Figures and Tables

**Figure 1 materials-13-04256-f001:**
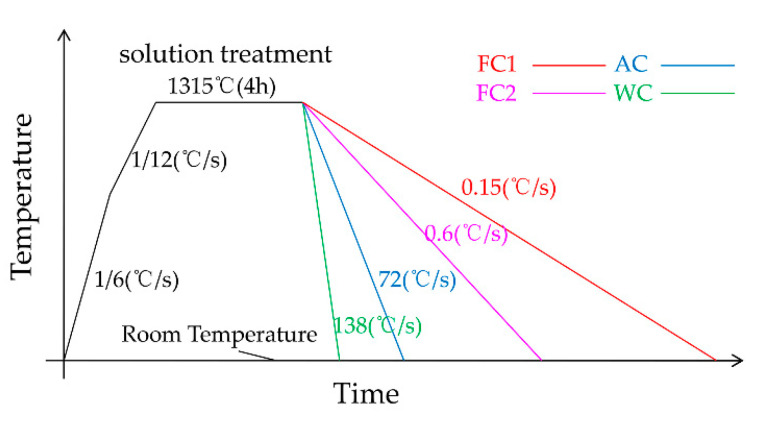
Temperature–time curve of four cooling rates.

**Figure 2 materials-13-04256-f002:**
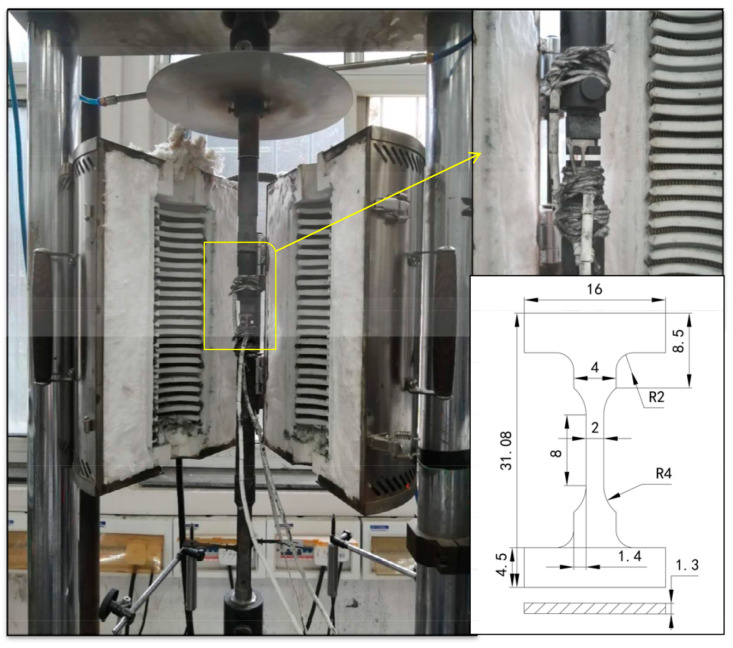
Dimensions and assembly drawings of I-shaped test pieces for tensile tests at 980 °C.

**Figure 3 materials-13-04256-f003:**
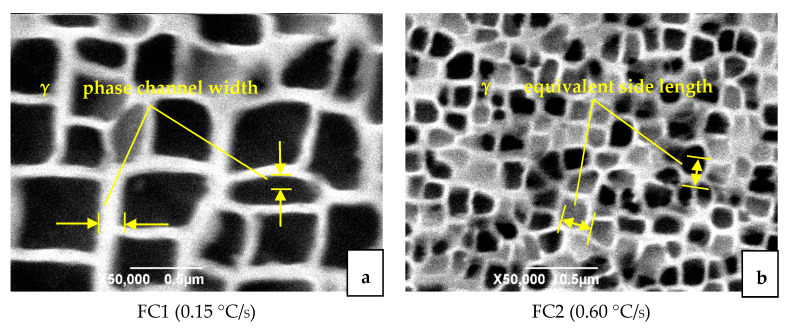
Microstructure of materials obtained by different cooling rates [23].

**Figure 4 materials-13-04256-f004:**
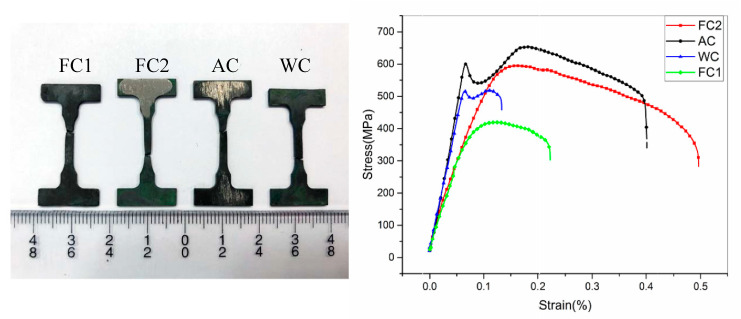
Test piece diagram and stress–strain curve after high-temperature (980 °C) tensile fracture.

**Figure 5 materials-13-04256-f005:**
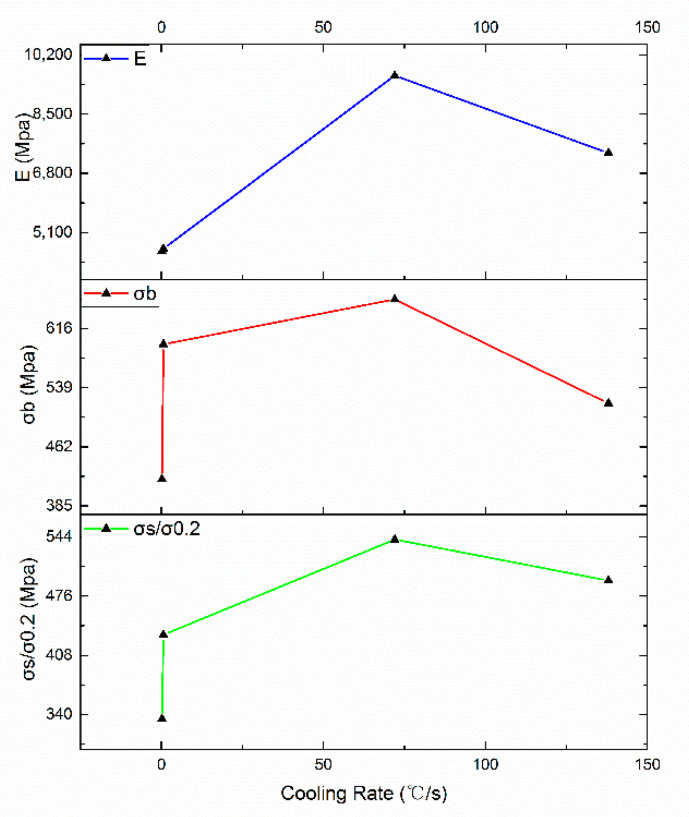
Yield strength (σ_s_/σ_0.2_), ultimate tensile strength (σ_b_) and elastic modulus (E) with different cooling rates.

**Figure 6 materials-13-04256-f006:**
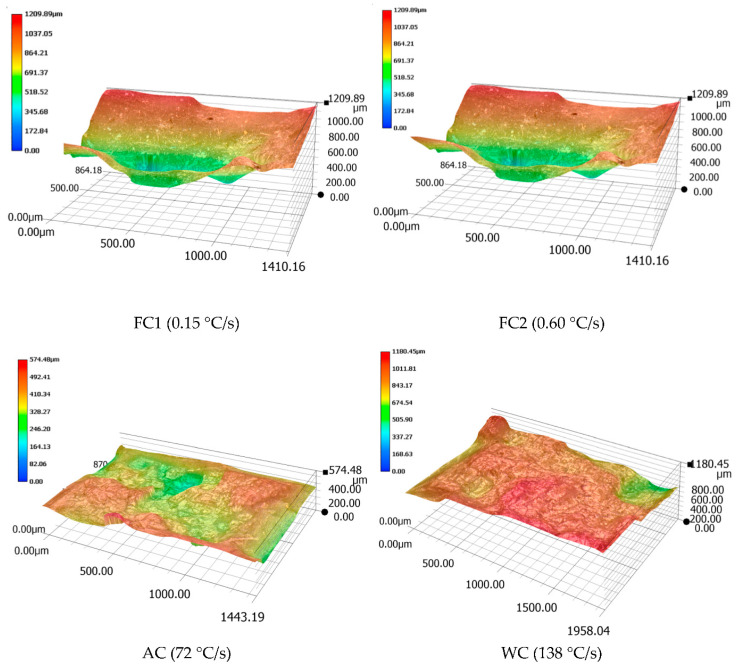
Contour map of the fracture surface of the samples after high-temperature tensile.

**Figure 7 materials-13-04256-f007:**
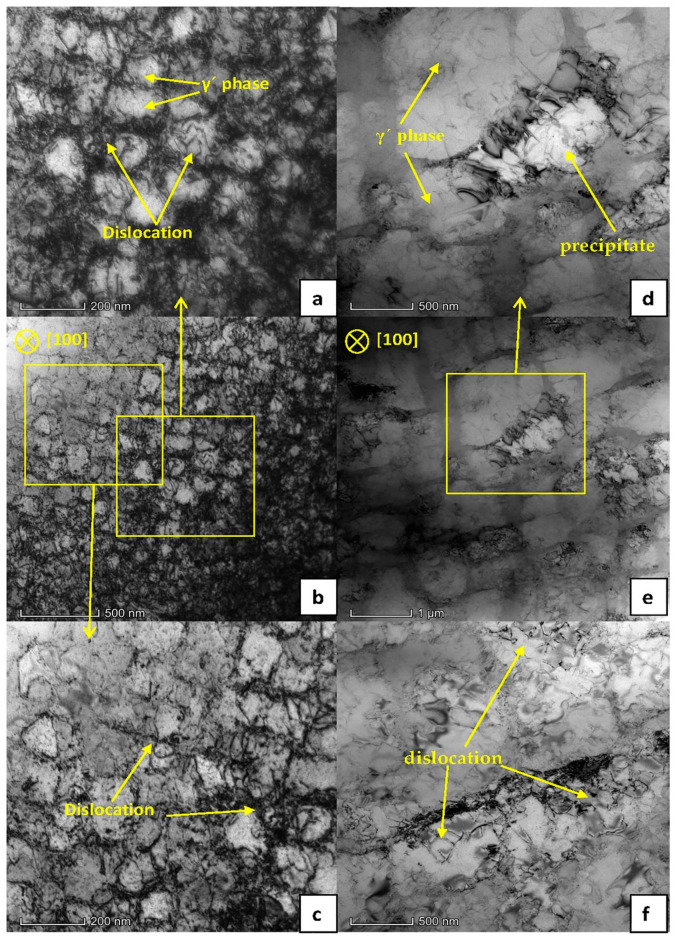
Bright-field TEM microstructure of the side of the fracture surface after tensile test at 980 °C. (**a**–**c**) Cooling rate is AC (72 °C/s); (**d**–**f**) cooling rate is FC1 (0.15 °C/S).

**Figure 8 materials-13-04256-f008:**
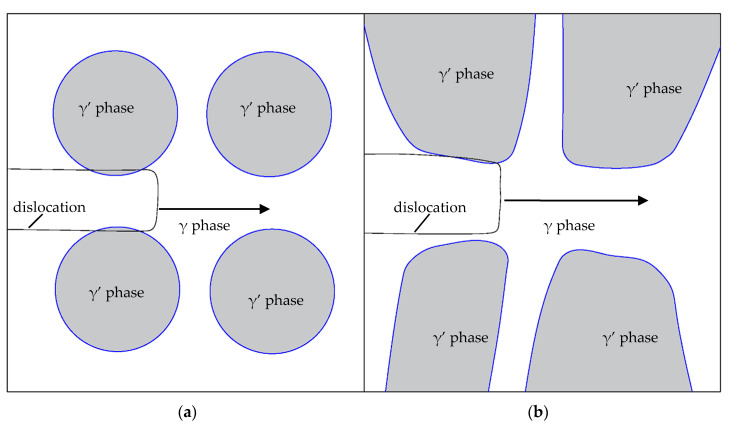
Schematic diagram of the passage of dislocations in (**a**) AC and (**b**) FC1 matrix-phase channels.

**Table 1 materials-13-04256-t001:** Nominal chemical composition of the nickel-based superalloy in wt%.

C	Cr	Ni	Co	W	Mo	Al	Ti	Ta	Re	Nb	B	Si	Hf
0.015	4.0	Bal.	9.0	8.0	2.0	5.7	≤0.10	7.0	2.2	1.0	≤0.02	≤0.02	1.0

**Table 2 materials-13-04256-t002:** Microstructure parameters under different heat treatment systems.

Number	Cooling Rate	γ′ Microstructure	γ′ Equivalent Side Length	γ′ Phase Channel Width
1	FC1 (0.15 °C/s)	Approximate cube	375 nm	30 nm
2	FC2 (0.60 °C/_S_)	Shape of irregular surface	183 nm	14 nm
3	AC (72 °C/_S_)	Approximate sphere	86 nm	8 nm
4	WC (138 °C/_S_)	Irregular dots	20 nm	1 nm

**Table 3 materials-13-04256-t003:** Tensile properties of nickel-based alloy materials with different cooling rates.

Number	Cooling Rate	Yield Strength (σ_s_/σ_0.2_)	Ultimate Tensile strength (σ_b_)	Elastic Modulus (E)	Elongation (δ)	Rate of Reduction in Area (Ψ)
1	FC1 (0.15 ℃/s)	334.88 MPa	419.73 MPa	4.568 GPa	41.66%	32.64%
2	FC2 (0.60 ℃/s)	431.42 MPa	595.41 MPa	4.634 GPa	22.23%	46.49%
3	AC (72 ℃/s)	540.95 MPa	653.91 MPa	9.603 GPa	40.15%	43.02%
4	WC (138 ℃/s)	493.74 MPa	518.61 MPa	7.381 GPa	13.32%	20.71%

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
