# Peer review of "The Effect of the Cooling Rates on the Microstructure and High-Temperature Mechanical Properties of a Nickel-Based Single Crystal Superalloy"

_materials, 2020, doi:10.3390/ma13194256_

Round 1
Reviewer 1 Report
Notes in the attachment.

Author Response
Thank you so much for your valuable comments.

Reviewer 2 Report
Dear Authors,
Your research is interesting and the experiments performed are accordingly selected.
Unfortunately, bad English obscure somewhat your discussions and conclusions, thus a English proof is mandatory.
Also, in the introduction several paragraphs introducing the metallurgical phases (structure, properties) would improve the article.
Here are some recommendations:
- in materials science the term used is microstructure, micromorphology is used frequently in biology and mineralogy.
Please use microstructure instead of micromorphology.
Line 12 - 14: revise paragraph, first sentence has no verb
Line 15: ... study the influence of different solid solution cooling rates on the... / ... study the influence of various cooling rates on the ...
Line 16 - 19: revise paragraph
Line 22: please rephrase
Line 31: solution treatment and aging treatment / solution and aging treatment
Line 36: instead of "reasonable heat treatment" use "adequate heat treatment" throughout the document
Line 60: Microstructure is capitalized and it should not be
Line 65: solution cooling rate - use simply "cooling rate" throughout the document
Line 71: width of the gamma phase channel - I think a definition would help the reader understand the meaning of the measurement
Line 75: ... high temperature mechanical properties were tested / high temperature mechanical properties were determined
Line 81: Lauie method? I think it's Laue's method
Line 86: the as-cast single crystal...
line 88-90: use the international symbolism, s for second, not S
line 95: 3x5x5mm is not a cube - please revise, and test blocks were cut
line 97: metallographic specimens were prepared for observation by ...
line 99: chemically corroded / etched
line 99-100: revise the etchant used
- also, the manufacturer and type of the equipment used should be mentioned in text
line 109-112: rephrase
line 118 and 120: high temperature stretching / high temperature tensile test
figure 1: use a common presentation for all graphs. Measurement units should be in brackets.
The representation of the heat treatment cycle is not adequate. A secondary temperature y axis should be used or a log scale on the time axis.
The whole thermal cycle is shown in fig. 1 and the cooling stage is not adequately represented - this should be your focus, given the figure title.
line 129: there ARE almost no carbides ...
line 136: rephrase
line 139-140: there are no results to support your statement on a statistical analysis (no mean value, no standard deviation or uncertainty associated with your results).
Line 144: rephrase
line 146: irregular square?
line 150: cooling rate is AC / cooling rate is 72deg C/s / when air cooled
line 153: furnace cold / furnace cooled
line 155: cooling rate is WC / when water cooling was used
line 157: rephrase
Table 2 - rephrase gamma prime shape description
line 174, 175: interface energy / surface energy
line 177: edge passivation - I do not understand the meaning. Rephrase.
Figure 4 - measurement units MPa not Mpa
line 183: tensile rate is 0.02mm/min / the test speed is ...
line 185: yield strength (lower yield strength) - I do not understand. Apparent and conventional yield strength?
line 187-187: air cooling and water cooling heat treatment ... - please rephrase.
line 188: i fail to understand what you mean by: the elastic deformation stage is longer.
line 203: have larger fractures - explain or rephrase
line 205: The material exhibits a certain degree o brittleness - rephrase or explain what you mean.
line 207-208: similar observations for certain degree of plasticity.
Table 3 - elastic modulus is in GPa, not Gpa
Also please check the values reported for the elastic modulus. I strongly doubt that it reaches such low values, lower than 10GPa.
For nickel single crystals the elastic modulus, depending on direction, ranges from 120-260GPa at room temperature and they can drop down to 90-100GPa at 800C.
Line 232 - 234: rephrase
Fig. 7.b - The feature that the steeper arrow is pointing to does not resemble to a dislocation (in my opinion). Perhaps the arrow head should be a little lower.
Line 238-240: please rephrase
Figure 8 - please elaborate the figure, ad arrows showing the direction and mark the dislocation in the figure
Author Response
Some replies are in the document, others have been revised in the manuscript. Thank you so much for your valuable comments.

Reviewer 3 Report
Please see enclosed file

Author Response

(The authors gave the same response as above.)

Round 2
Reviewer 1 Report
The authors made corrections according to my comments. The corrections made are satisfactory.